# *TaeC*: A manually annotated text dataset for trait and phenotype extraction and entity linking in wheat breeding literature

**Claire Nédellec**[1]*, **Clara Sauvion**[1], **Robert Bossy**[1], **Mariya Borovikova**[1,2],
**Louise Deléger**[1]

**1** Université Paris-Saclay, INRAE, MaIAGE, Jouy-en-Josas, France, **2** TETIS, Univ. Montpellier, AgroParisTech, CIRAD, CNRS, INRAE, Montpellier, France

* claire.nedellec@inrae.fr

**Data Availability Statement:** - The corpus dataset TaeC is available under CC-BY-ND License at: https://entrepot.recherche.data.gouv.fr/dataset.xhtml?persistentId=doi:10.57745/GCYGQ - The

## Abstract

Wheat varieties show a large diversity of traits and phenotypes. Linking them to genetic variability is essential for shorter and more efficient wheat breeding programs. A growing number of plant molecular information networks provide interlinked interoperable data to support the discovery of gene-phenotype interactions. A large body of scientific literature and observational data obtained in-field and under controlled conditions document wheat breeding experiments. The cross-referencing of this complementary information is essential. Text from databases and scientific publications has been identified early on as a relevant source of information. However, the wide variety of terms used to refer to traits and phenotype values makes it difficult to find and cross-reference the textual information, e.g. simple dictionary lookup methods miss relevant terms. Corpora with manually annotated examples are thus needed to evaluate and train textual information extraction methods. While several corpora contain annotations of human and animal phenotypes, no corpus is available for plant traits. This hinders the evaluation of text mining-based crop knowledge graphs (e.g. AgroLD, KnetMiner, WheatIS-FAIDARE) and limits the ability to train machine learning methods and improve the quality of information. The *Triticum aestivum trait Corpus* is a new gold standard for traits and phenotypes of wheat. It consists of 528 PubMed references that are fully annotated by trait, phenotype, and species. We address the interoperability challenge of crossing sparse assay data and publications by using the *Wheat Trait and Phenotype Ontology* to normalize trait mentions and the species taxonomy of the *National Center for Biotechnology Information* to normalize species. The paper describes the construction of the corpus. A study of the performance of state-of-the-art language models for both named entity recognition and linking tasks trained on the corpus shows that it is suitable for training and evaluation. This corpus is currently the most comprehensive manually annotated corpus for natural language processing studies on crop phenotype information from the literature.

Wheat Trait and Phenotype Ontology (WTO) is available under CC-BY-ND License at http://agroportal.lirmm.fr/ontologies/WHEATPHENOTYPE) - The AlvisNLP bread wheat workflow software is available under Apache License at https://forgemia.inra.fr/migale/wheat-tm - The code of the ToMap method is available under Apache License at https://github.com/Bibliome/alvisnlp/tree/master/alvisnlp-bibliome/src/main/java/fr/inra/maiage/bibliome/alvisnlp/bibliomefactory/modules/tomap.

**Funding:** The grant ANR-18-CE23-0017 (D2KAB project) is the funding source for this work and we confirm that the authors received funding for this work. The funders had no role in study design, data collection and analysis, decision to publish, or preparation of the manuscript.

**Competing interests:** The authors have declared that no competing interests exist.

## Introduction

The improvement of agricultural crop varieties has become an international concern due to the increasing demand for food to support a growing world population and the need to counteract the decline of resources, especially water and oil. This is especially true for wheat, the most widely grown crop after rice. Climate change and reduction in inputs (water, fertilizers, and pesticides) and acreages are new environmental constraints that lead to an increased need for wheat variety traits related to tolerance to water limitation and temperature stress, resistance to wind (e.g., stalk mechanical strength, resistance to lodging) and disease, or nutrient use efficiency [1]. Wheat is used in a wide range of new food products, the demand for which is increasing due to changes in dietary habits and the search for health benefits. Wheat grain composition is therefore an important target for change: for example, low gluten preparations, textured proteins for vegetarian and low-carbohydrate products, starch composition involved in the preparation of baked goods, meat products and confectionery, and lower levels of synthetic chemical products in the design of breading, coating, and brine additive. Therefore, a wide variety of wheat traits are being investigated, ranging from response to environmental conditions (biotic and abiotic), quality (for milling, for food), growth (e.g., yield, vigor, nutrient use), morphology, to reproduction.

The recent advent of genomic tools contributed to improving the linkage between molecular markers and genes of agronomic interest. This information is being integrated into increasingly shorter breeding programs to move from genetic toward genomic variety selection. Recently, many varieties and molecular markers have been developed for bread wheat [2]. Data from breeding experiments are spread over thousands of heterogeneous datasets and publications [3].

Given the heterogeneity and sheer volume of genetic and phenotypic information, a number of national and international initiatives have been launched to collect, aggregate and make interoperable information on crop species in knowledge graphs to facilitate the discovery of complex phenotype-genotype associations and the understanding of the underlying biological mechanisms. They develop new semantic web-based tools for the semantic description of agricultural data, making them actionable and openly accessible, according to the FAIR (findable, accessible, interoperable, and reusable) principles [4]. The most prominent collaborative efforts, KnetMiner [5], WheatIS data discovery tool [6], FAIDARE FAIR Data-finder for Agricultural REsearch [7] or AgroLD-Southgreen [8, 9] uses ontologies of traits and phenotypes, e.g. Plant Trait Ontology [10] and Crop Ontology [11], to describe phenotypic traits in plants by controlled vocabularies and make them interoperable.

Although their primary phenotypic data sources are phenotypic assay datasets, they acknowledge the importance of scientific publications and textual information. The identification of the trait and phenotypic expressions in free text, i.e. named entity recognition (NER), and their association with ontology classes, i.e. named entity linking (NEL) or normalization, are two complex tasks because of the large number of traits and the broad diversity of the vocabulary used to designate them. The KnetMiner's text analysis pipeline parses large volumes of scientific literature abstracts to identify phenotypic trait descriptions and normalizes them by dictionary-based lookup of ontology class labels, primarily from Plant Trait Ontology and Gene Ontology, and computes sentence-based co-occurrence [12]. WheatIS provides trait cross-references obtained by indexing phenotypic assay datasets and by semantic parsing PubMed references through the AlvisNLP pipeline [13]. AgroLD team is currently developing methods to extract information embedded in scientific publications [9]. The text mining results of KnetMiner or WheatIS are illustrated by examples without a description of the general assessment of their effectiveness [5, 13]. Traditionally, the ability of information extraction

methods to recognize entities and to predict their correct class is evaluated by comparing their predictions to the manual annotations of corpora, i.e. gold standard.

Moreover, the state-of-the-art algorithms for NER and NEL heavily rely on supervised Machine Learning methods. Although the number of training examples needed to train machine learning-based methods decreases with the advent of few-shot and zero-shot algorithms [14], the quality of algorithm prediction remains correlated with the availability of training examples of the target information. To the best of our knowledge, there is no such corpus that could be used to evaluate and train the methods in this domain.

To fill this gap, we have developed the *Triticum aestivum trait Corpus* (*TaeC*). The entity types of *TaeC* include trait, phenotype, and species. Traits are observable characteristics such as the height of the plant or the resistance to a particular disease. The phenotypes are the values of the traits, *e.g.*, *1,2 m* as the *height value*, or *highly resistant to wheat blast* as the value for the *wheat blast resistance* trait.

This work was carried out as part of the D2KAB project (*Data to Knowledge in Agriculture and Biodiversity*; https://d2kab.mystrikingly.com/) with the aim of providing the plant data discovery federation (FAIDARE WheatIS) with a formal way to evaluate and train its underlying text mining methods and contribute to its quality measure. NER and NEL corpora in Life Science mainly focus on human health. A few contain annotations of plant properties, but none provide annotations of traits with an ontology. We used the *Wheat Trait and Phenotype Ontology* (WTO; http://agroportal.lirmm.fr/ontologies/WHEATPHENOTYPE) [15] as a reference resource for textual annotation following the work of the OpenMinTeD project for WheatIS (https://www.fosteropenscience.eu/node/2316). Species are relevant entities here because many hybrid varieties obtained by crossing wild (e.g., *Aegilops tauschii*, an annual grass species) and cultivated plants (e.g., *Hordeum vulgare*, barley) are mentioned besides bread wheat (*Triticum aestivum*), durum wheat (*Triticum durum*), and their subspecies. We used the NCBI taxonomy (https://www.ncbi.nlm.nih.gov/taxonomy) as the reference resource for species standard annotation. The NCBI taxonomy is a worldwide recognized resource that is also relevant to the association of genes to phenotypes through its linked genetic sequence databases.

The paper presents the context and the rationale for the design of *TaeC*. It details its construction steps from the document selection to the manual annotation. It discusses the performance of state-of-the-art methods applied to *TaeC* to demonstrate its relevance for the future evaluation and training of text analysis pipelines embedded in crop information networks.

## Background

First, we surveyed existing corpora to determine if we could reuse some annotations four our information extraction purpose of phenotypic traits. Most work has focused on human and animal phenotypes and their linkage with genetic peculiarities and abnormalities. The popular *Phenotype-Gene Relations* corpus (PGR) [16] is annotated with the *Human Phenotype Ontology* (HPO), a standard vocabulary for phenotypic abnormalities that occur in human diseases [17], using the machine learning-based NER tool IHP (*Identifying Human Phenotypes*) [18]. HPO does not include regular traits such as eye color but only abnormal traits such as ocular albinism and is limited to humans and animals, making it irrelevant to plants. The Bacteria Biotope corpus [19] is annotated using the *OntoBiotope* ontology [20], a standard vocabulary for biotopes and phenotypes of microbes. Although *Ontobiotope* describes normal phenotypes, microbes are too different from plants for the corpus and ontology to be reusable, even if some of the highest-level classes are relevant, such as biotic and abiotic stress response or morphology.

Plant biology has so far been relatively underrepresented as a topic in natural language processing for the biological and medical domains (BioNLP]. For *Arabidopsis thaliana*, there have been some recent information extraction initiatives, such as the *SeeDev* reference corpus [21] and the *KnownLeaf* literature curation system [22]. The *SeeDev* corpus concentrates on seed development described at the molecular level, which is different from our subject. The *Known-leaf* corpus focuses on the regulatory mechanisms of leaf growth and development, and key genes related to relevant mutant phenotypes. Our goal differs from that of *Knownleaf* in that we aim to account for the full diversity of trait values in wheat varieties and we do not consider wheat phenotypes as normal or abnormal with respect to a normal genetic reference. Moreover, plant traits in the *Knownleaf* corpus follow the entity-attribute-value (EAV) model: they are described using terms from the *Phenotype*, *Attribute*, *and Trait Ontology* (PATO) [23] that formally combines plant parts and tissues from the *Plant Trait Ontology* (TO) [24], and the *Brenda Tissue Ontology* [25]. For example, in the text, "The reduced leaf area in the hub1-1 mutant", the *reduced leaf area* phenotype is formalized as three distinct entities, *area* as the property, *reduced* as the value, and *leaf* as the plant part. For *TaeC*, our modeling choice was different, and we preferred textual annotations that encompass the plant part, the trait, and possibly its value rather than a formal distinction of the entities as the *Knownleaf* project does. Indeed, the goal of the *D2KAB* project is to facilitate the retrieval and reading of information for human users by using expressions that are as close as possible to their designation habits.

The annotation of the *D2KAB* wheat data is thus done by ontologies that meet this need: the Wheat Trait and Phenotype Ontology (WTO) for text data and the *Wheat Crop Ontology* CO_321 for experimental data. In both ontologies, the trait classes gather the trait and the plant part that it characterizes in single names; for example, *grain color* is the label of WTO:0000141 and CO_321:0000037 trait classes.

Most of the corpora with species annotations are related to biomedicine, although some corpora also include plant species. LINNAEUS [26], COPIOUS [27], and S1000 [28] are the most prominent among them. Most species of the LINNAEUS corpus reflect the PubMed source focus and are related to human health, model species, and microbes. COPIOUS documents from the *Biodiversity Heritage Library* include botanical species limited to the Philippine biodiversity. The S1000 corpus is dedicated to biodiversity surveys. Its botanical subset contains 125 scientific articles with 357 plant species annotations, among which only a dozen species are relevant to wheat breeding, e.g., *Zea mays*, *Triticum aestivum*, *Oryza sativa*. The lack of a relevant corpus for text mining studies on wheat phenotypic traits was then the motivation for our work on *TaeC*.

## Materials and methods

In this section, we describe how we constructed the *TaeC* gold standard. We present our method for selecting documents, the annotation schema, the annotation guidelines, and the annotation process. In order to assess the suitability of the corpus for the development of the NER and NEL methods, we selected the information extraction methods described in this section, which we then applied to the corpus.

### Document selection

We selected PubMed (https://www.ncbi.nlm.nih.gov/pmc/) as the bibliographic source for scientific documents because it is fully open, the titles and abstracts of references are short and focused texts, and they contain a wealth of trait and phenotype mentions. The documents of *TaeC* were selected using the query in Table 1.

**Table 1. Corpus document selection query to PubMed with wheat breeding, genetics and species criteria.**

1. *(("biomarkers"[MeSH Terms] OR "biomarkers"[All Fields] OR "marker"[All Fields] OR "markers"[All Fields])*
2. *AND ("genes"[MeSH Terms] OR "genes"[All Fields] OR "gene"[All Fields])*
3. *AND ("triticum"[MeSH Terms] OR "triticum"[All Fields] OR "wheat"[All Fields] OR "wheat s"[All Fields] OR "wheats"[All Fields]))*
4. *AND ((fha[Filter]) AND (english[Filter]))*

The first two parts of the query, with the keywords marker and gene, select documents about wheat breeding and exclude documents about other uses of wheat, such as food processing or food composition. The third part selects documents about wheat species. The last part selects documents that are in English and contain an abstract.

The query performed on PubMed on April 22, 2022, retrieved 5,596 documents, mostly published after 2011. The distribution of the document set is biased towards resistance to disease, while our goal is to build a corpus that is representative of the trait diversity. To select a representative subset to be annotated, we applied the *AlvisNLP* workflow dedicated to bread wheat [13]. It automatically annotates the trait entities in titles and abstracts and link them to the *Wheat Trait and Phenotype Ontology* (WTO) classes [15]. Using these rough class annotations, we were able to generate a subcorpus of 528 documents with an even distribution of the topics. WTO contains six main topics (i.e. subtrees), development, growth, morphology, quality, reproduction, and response to environmental conditions. We generated 90 subsets of 6 documents where the selection is biased so that each set contains at least one document with entities annotated by one of the topics, i.e. at least one class of a subtree, resulting in a uniform thematic distribution among the documents. Of course, it may occur that a document is annotated by more than one topic.

## Annotation schema

The annotation scheme consists of three entity types: Species, Trait and Phenotype. Species are normalized by the NCBI taxonomy. We have chosen this reference because it is very widely used for the indexing of the genetic and biological properties of crop species.

We preferred the *Wheat Trait and Phenotype Ontology* (WTO) for annotating Trait and Phenotype entity types over the *Wheat Crop Ontology CO_321* because it was designed to meet the requirements of annotating textual data with ontologies [15], while *CO_321* was designed for the description of phenotyping assays [11]. Indeed, the description of the plant properties differs depending on whether they are found in scientific texts or experimental observation datasets. Traits mentioned in documents usually pertain to the general properties of the wheat variety. On the other hand, observational data describe specific states of the plant within a limited spatial and temporal scope which must be aggregated and experimentally confirmed to derive the general characteristics of the observed wheat variety.

WTO contains two parts, the Environmental Condition subtree and the Plant Property subtree, which is further subdivided into the Trait subtree (WTO_0000006) and the Phenotype subtree (WTO_0000005), for a total of 596 classes covering all plant property dimensions. The classes of these two subtrees are respectively used to annotate the Trait and Phenotype entities of the corpus documents.

Figs 1 and 2 show illustrative annotation examples on text excerpts from the *TaeC* corpus which have been kept short for the sake of readability. Fig 1 shows a simple annotation example of text where *earliness* is an entity of trait type, and *wheat* is an entity of species type.

Fig 2 shows a more complex example where the entity of the trait type is the long-term *resistance to single isolates of M. graminicola*. It is associated with the WTO_0000554 class

**Fig 1. Example of annotation of single-word trait and wheat species in a text excerpt from the *TaeC* corpus.** The WTO class associated with the *earliness* term in the text is *plant precocity* (WTO_0000100). The NCBI taxonomy class associated with the *wheat* term in the text is *Triticum aestivum* (TaxID_4565).

*Resistance to Septoria leaf blotch*. This example illustrates the case where the entity name is not a short and simple term that is easy to identify. Even more complex is the association of the entity with the class, which requires an expert understanding that *M. graminicola* is a causal agent of the Septoria leaf blotch disease of wheat.

## Annotation guidelines

We have written annotation guidelines that define the entities, give examples and counter-examples of annotations, and detail exceptions and borderline cases. We adapted the annotation guidelines of [29] to the tasks: we removed sections about irrelevant relationships and entities, and added new sections for entity linking of traits, phenotypes, and species. This resulted in an eight-page document [30].

## Annotation process and tools

The corpus with entities pre-annotated by the *AlvisNLP* workflow for bread wheat was made available to the experts to save time. Two annotators successively annotated 528 documents; the first annotator had previous experience in manually annotating biomedical documents and performed an extensive annotation, the second annotator was an expert in biology and thoroughly reviewed all annotations. Annotation issues were addressed in collaboration with two textual annotation experts and two experts in wheat agronomic traits and disease resistance, which were the most complex traits to annotate.

For manual text annotation, we used the *AlvisAE* editor [31], which we have successfully used in previous annotation campaigns involving entity normalization using ontologies [19]. It provides a graphical interface and it is implemented as a Web application, thus facilitating the participation and collaboration by domain experts. We created an annotation campaign for the *TaeC* project. Fig 3 shows a screenshot of the main window of the *AlvisAE* instance for *TaeC*.

*Parents and DH progeny were tested for resistance to single isolates of M. graminicola in a growth chamber at the seedling stage, and to an isolate mixture at the adult plant stage, in field trials.*

**Fig 2. Example of annotation of complex trait mention in a text excerpt from the *TaeC* corpus.**

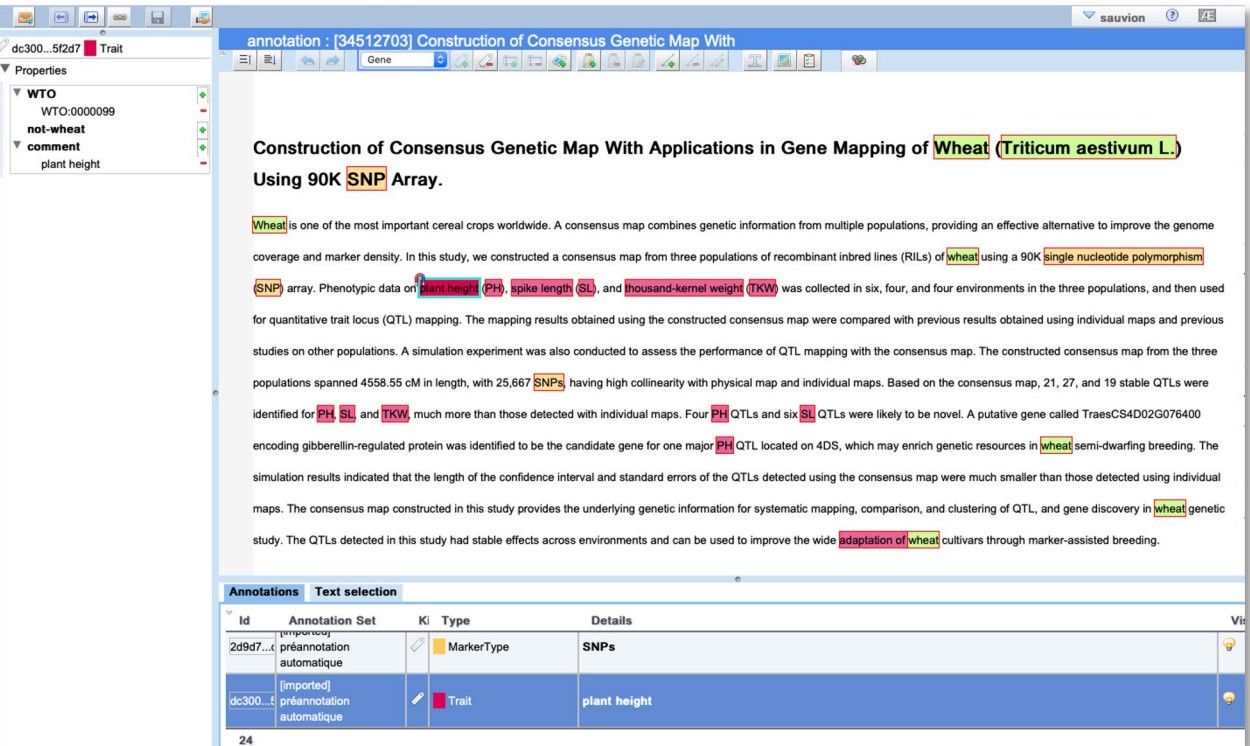

**Fig 3. Screenshot of the *AlvisAE* annotation editor used to manually annotate *TaeC* with many different annotations.**

Automatic consistency checking was regularly performed to ensure high-quality annotation. The rules to be checked were (1) all entity mentions are tagged by classes, (2) the same classes tag identical entity mentions, and (3) there is no entity with both annotated and unannotated occurrences. The rules may not be relevant in some cases, i.e., a given expression may have different meanings depending on the context. The consistency check was therefore intended to warn the annotators but not to be used as an automatic revision tool.

## Information extraction methods

In order to evaluate the suitability of the corpus and to ensure that the information extraction tasks are feasible but not trivial, we have selected a few named entity recognition and linking methods. Our goal at this stage is to assess the complexity of the tasks and to identify critical issues by testing some relevant methods; it is not to build an inference text analysis pipeline to feed WheatIS with textual information that is to be left for further work.

We considered rule-based and machine learning methods for named entity recognition and linking in *TaeC* and we explored an end-to-end solution by applying the best NEL model to the predictions of the best NER model.

**Rule-based methods.**   Previous work on the recognition and linking of wheat phenotype entities in texts includes the application of the ToMap method [32] using WTO [15]. The preliminary experimental results published in [13] on the application of ToMap to the recognition and linking of wheat phenotype entities in text were encouraging. We therefore used it for the joint NER and NEL tasks on TaeC. The ToMap method applies to candidate entities extracted from texts by the YaTeA term extractor [33]. It computes the similarity of candidate entities and candidate class labels based on their syntactic structure and common words in the same

spirit as MetaMap [34]. The syntactic structure information is especially useful for the processing of complex prepositional phrases, which are common in the wheat breeding literature; for example, the phrase *Fusarium head blight resistance* should be mapped to the class WTO: 0000483 *resistance to Fusarium head blight*. ToMap computes the correspondence between their main syntactic chunks [resistance [to [Fusarium head blight]]] / [[Fusarium head blight] resistance] that are the similar (i.e. same head, same adjunct), while their surface forms differ.

**Machine learning-based methods.**   Recent successful NER and NEL machine learning methods are based on neural networks and language models. The recognition and linking tasks are both treated as classification tasks. Common NER approaches start from a general language model that is fine-tuned to the particular task using training examples. One prominent example is the RoBERTa model [35], which has become widely used due to its ability to be fine-tuned for multiple tasks, including NER [36]. In the biological domain, the BioBERT model [37] is used for both NER [38, 39] and NEL [40] tasks with an additional fine-tuning stage. In our experiments, we employ the classic fine-tuning approach applied to RoBERTa and BioBERT models, with the addition of a dense classification layer and a softmax function, as introduced in [41] for classification with BERT language model.

For NEL on biomedical data, machine learning approaches combined with semantic ontology representations are prevalent. For the NEL task involving traits and phenotypes, we use the C-Norm and BioSyn algorithms. The C-Norm method [42] achieves state-of-the-art performance on the Bacteria Biotope dataset, which has good similarities to *TaeC*, i.e., deep ontology and complex entity terms. C-Norm represents terms in the texts using Word2vec embeddings [43], and it represents ontology classes using vectors that integrate hierarchical information from the ontology [44]. It combines a single-layer feedforward neural network and a shallow convolutional neural network.

Another notable NEL algorithm in the biomedical domain is BioSyn [45]. The authors of this method train a dense entity representation space using the BioBERT model. It uses synonym marginalization techniques as the objective function for training and a similarity function, leveraging top-k similar candidates to update model parameters iteratively.

**Species recognition and linking.**   Species recognition has inspired a significant amount of work. Most of the methods are dictionary-based, enhanced by additional local variation rules (e.g., Levenstein distance), contextual rules or intrinsic document word disambiguation, although there is growing interest in machine learning-based solutions; see [39] for a survey. The limited range of species in the corpus does not require such advanced tools for their recognition and linking as in the general open domain.

The NEL task for *TaeC* is to link the entities to the NCBI taxonomy, so we used this taxonomy as a dictionary for predicting and linking the entities. We chose the *AlvisTaxa* tool because of its good performance in the NER and NEL microbial species [46]. It uses rewriting rules to generate species name variations, including abbreviations and acronyms, and performs contextual disambiguation. We found the method appropriate given that species names occurring in *TaeC* are both scientific and vernacular names and include abbreviations, and the method achieved good performances in Life Science tasks [19].

**Methods and parameters.**   Both the rule-based *AlvisTaxa* and *ToMap* algorithms are part of the *AlvisNLP* workflow dedicated to bread wheat that we used for the pre-annotation. The AlvisNLP bread wheat workflow is available at: https://forgemia.inra.fr/migale/wheat-tm. It includes the wheat-specific lexica used by ToMap.

The code of the ToMap method is available at https://github.com/Bibliome/alvisnlp/tree/master/alvisnlp-bibliome/src/main/java/fr/inra/maiage/bibliome/alvisnlp/bibliomefactory/modules/tomap

C-Norm and BioSyn were run with the default parameters provided by the authors.

The values of the hyperparameters of RoBERTa, BioBERT, BioSyn and C-Norm were as follows:

- Number of training epoch: 10

- Learning rate: 5e5

- Gradient accumulation step: 2

- Random seed: -42

- Maximum gradient norm: 10

## Results

This section presents the details of *TaeC* annotations, and the NER/NEL results that we obtained from our experiments.

### *TaeC* corpus

Table 2 shows entity annotation and class frequencies of the corpus per entity type.

The high density of 13 trait and phenotype annotations per document confirms the relevant selection of PubMed bibliographic database as a source of dense descriptions. The diversity of the mentions is shown by the high number of unique expressions compared to the number of occurrences. Each phenotype mention is repeated 1.8 times on average (1607/873), and each trait mention is repeated 3.6 times on average (5477/1508).

Trait and phenotype entities are tagged by 232 different classes, yielding 10.2 unique mentions and 30.5 occurrences per class on average, which is high. Among 5477 Trait occurrences, ~35% (1901) are equal to their class label and among 1607 Phenotype occurrences, only ~3% (45) are equal to their class label. This makes the annotation of Phenotype occurrences much more complex.

The diversity of the WTO classes used demonstrates the efficiency of the document sampling.

As might be expected, the distribution varies significantly by class; see examples (1) and (2).

1. the WTO:0000072 *class culm length* has four forms in the corpus, *culm length*, *stem length*, and their CL and SL acronyms,

2. while WTO:0000146 *grain protein content* presents 45 different forms, e.g. *seed protein content*, *protein contents*, *reduction in protein body number*, *reduced grain protein content*.

The diversity of taxon names is lower, with 2.6 forms per class on average. The number of occurrences (7 per document) is surprisingly high, but the occurrences are highly repeated (14 times on average), especially the word *wheat* with 1,595 occurrences.

**Table 2. Figures of the *TaeC* corpus in number of annotated entities and classes per type.**

| Manually annotated documents | 528 | | |
|---|---|---|---|
| Tokens | 138,274 | 262 per document | |
| Occurrences of phenotype | 1,607 | 3.04 per document | 873 unique |
| Occurrences of trait | 5,477 | 10.4 per document | 1,508 unique |
| Classes of traits or phenotype | 232 | 10.2 forms per class | |
| Occurrences of plant species | 3,731 | 7 per document | 266 unique |
| Classes of taxon | 104 | 2.6 forms per class | |
| Total occurrences | 10,815 | | |

**Trait and phenotype annotation.** The trait and phenotype entities are adjectival or nominal expressions. Noun phrases are the most frequent, but adjectives also frequently express phenotypes (e.g., *sprouting-resistant*, *salt-tolerant*).

Many of them are acronyms (e.g., *GPC* standing for *Grain Protein Content*); prepositional phrases are common (e.g., *ratio of the quantity of glutenin to those of gliadin*, *number of grains per spike*); some forms include parenthesized expressions (e.g., *number of flowering branches (spikelets) per node*) where the term into parenthesis can be an acronym (e.g., *Efficient phosphate (Pi) uptake*, *responses to low red light/far-red light (R/FR) ratios*). Discontinuous entities are rare. For instance, in the text <u>resistant to</u> both <u>WSMV</u> and <u>Triticum mosaic virus</u>, two discontinuous annotations of phenotypes are made, i.e., *resistant to WSMV*, and *resistant to* Triticum *mosaic virus*.

The entities differ from the class labels in many ways as shown in Table 3, requiring deep domain expertise.

Many acronyms are used (see examples (1) where *HTAP* stands for *high-temperature adult-plant* and (2)). Morphological and syntactic variations are frequent (see examples (3) and (4)). Synonyms are often used, for example, the plant parts in trait and phenotype names may be referred to by different names (e.g., *grain*/*kernel*/*seed*; *spike*/*ear; culm*/*stem*). WTO defines alternative names for those traits but not all.

The resistance to diseases can be expressed as the resistance to the pathogen agent as in examples (5) to (7). Fungi are the major cause of disease in wheat. Beyond their official names, alternative names for separate morphs, teleomorph, anamorph, and holomorph, are used. WTO records many of them. For instance, the eyes spot disease is caused by *Helgardia herpotrichoides* (syn *Pseudocercosporella herpotrichoides*, *Ramulispora herpotrichoides*, *Tapesia yallundae*, Cercosporella herpotrichoides). The corresponding resistance trait class WTO:0000482 *resistance to Eyesspot* has therefore five alternative names accordingly.

**Table 3. Examples of the wheat phenotype and Trait ontology labels and the corresponding text mentions in publications.**

|  | Text entity | Class label |
|---|---|---|
| (1) | *HTAP resistance* | **high-temperature resistance** |
| (2) | *TKW* | **thousand kernel weight** |
| (3) | *phenolic content* | **grain polyphenol content** |
| (4) | *number of tillers* | **shoot number per plant** |
| (5) | *resistance to Magnaporthe grisea* | **resistance to wheat blast** |
| (6) | *highly resistant to leaf rust caused by* **Puccinia triticina** | **resistance to Leaf Rust** |
| (7) | *resistance to single isolates of* **M. graminicola** | **resistance to Septoria Leaf Blotch** |
| (8) | *heading time* | **ear emergence time** |
| (9) | *number of fertile florets at anthesis* | **fertility** |
| (10) | *low molecular weight glutenin subunit* | **glutenin content** |
| (11) | *Resistance to the disease septoria tritici blotch caused by the fungus Mycosphaerella graminicola* | **resistance to Septoria Leaf Blotch** |
| (12) | *resistance against a number of other important P. graminis f. sp. tritici pathotypes* | **resistance to Stem rust** |
| (13) | *impact on frost damage in cereal reproductive tissues by influencing accumulation of genuine tolerance* | **frost tolerance** |
| (14) | *level of physical attachment of glumes to the rachilla of a spikelet* stands for. | **glume tenacity** |
| (15) | *grain yields of evaluated cultivars growing in the field under water-limited conditions* | **drought tolerance** |

The number of alternative names increased with the number of pathogen agents for a given disease, and the knowledge about causal agents evolves over time. Experts may disagree on the causal agents of diseases, and species distinction. For instance, the synonyms of WTO:0000510 *resistance to wheat blast* include *resistance to Magnaporthe grisea*, *resistance to Magnaporthe oryzae*, and *resistance to Pyricularia grisea* which are considered different species by some experts, notably NCBI. The contribution to *TaeC* annotation of experts in the fungi diseases of wheat has been significant in handling this issue. Other semantic variations are also frequent as shown by examples (8) to (10).

Long noun phrases with subordinate clauses, infinitives, participles, *gerunds*, and verbless clauses are hopefully rare but more frequent than in other corpora. (11), (12), and (13) are illustrative examples. The *TaeC* corpus includes paraphrases that reword traits and phenotypes in longer forms to make the meaning clearer and more precise. Examples of paraphrase are (14) and (15).

The confusion between the trait and the method used to measure its value is another source of ambiguity. For example, grain weight traits are usually qualified by the method, e.g., *thousand grain weight*, *test weight*. In this case, we include the name of the method in the annotation span. More complex cases are expressions such as *Zeleny sedimentation value*, mostly used in place of the trait name, WTO:0000613 *flour sedimentation volume* (related to *grain protein content*). The decision was also to annotate them as trait entities.

The annotation of recent articles in the corpus revealed the evolution of the domain due to the phenotyping method advances. New traits are studied, and new terms for denoting them occur. Examples of such new topics are related to plant architecture (tillers and root), physiology (hormone response, source-sink relationship), and reproduction. We extended WTO with 67 new specific classes to reflect the domain evolution. This new version is available at http://agroportal.lirmm.fr/ontologies/WHEATPHENOTYPE.

Consistency checking reveals manual annotation errors, among which the most frequent ones were over-specific or over-general class annotations of infrequent expressions because the size of WTO is too large for the experts to remember them all. These errors were automatically detected and manually corrected.

**Taxon annotation.**   Species are equally designated by their scientific name (*e.g.*, *Triticum aestivum* L.), sometimes abbreviated (*e.g.*, *T. aestivum*), and by their vernacular name (*e.g.*, bread wheat, barley, rice). Vernacular names can be ambiguous, especially *bread wheat*, which is frequently wrongly called *wheat*, although *wheat* is the vernacular general name of the *Triticum* genus that includes 19 species, not only *Triticum aestivum* species.

**TaeC format and distribution.**   As usual for annotated corpora, *TaeC* distribution includes (1) the text itself and its bibliographic metadata, (2) the entities, their position, and their type, and (3) the identifier of the ontology class.

It is provided in the BioNLP-ST standoff annotation format [47], which is suitable for representing entities linked to a semantic reference (Table 4). The text, the entities, and the semantic references are in separate files. The files are linked through the name of the file which plays the role of identifier.

The annotation set is split into three subsets, a train, a development, and a test subset with similar entity class distributions. The train and development subsets are made public. The test set will be used in a future shared task and remains hidden to prevent overtraining. *TaeC* is available under CC-BY-ND License at: https://entrepot.recherche.data.gouv.fr/dataset.xhtml?persistentId=doi:10.57745/GCYG3Q.

## Named entity recognition and linking

We evaluated the performance of state-of-the-art rule-based and machine learning methods on the *TaeC* corpus with the goal of estimating the difficulty of the NER and NEL tasks. We

**Table 4. File.txt contains the text of the document.** File.a1 contains the named entity annotations, i.e. the internal identifiers of the named entity, their type (i.e. Trait or Species), their position and their text form. File.a2 contains the semantic annotations, i.e. the internal identifier of the semantic annotation, the name of the reference (i.e. NCBI taxonomy, or WTO), the identifier of the named entity as declared in File.a1, and the external identifier of the class in the reference.

| |
|---|
| **File.txt** |
| *Efficiently tracking selection in a multiparental population*: *the case of earliness in wheat.* |
| **File.a1** |
| T1 Trait 75 83 earliness |
| T2 Species 88 92 wheat |
| **File.a2** |
| N1 NCBI_Taxonomy Annotation:T1 Referent: 4565 |
| N3 WTO Annotation:T2 Referent:WTO: 0000100 |

used the same metrics as the BioNLP Shared Task [47]. In addition to the usual recall and precision metrics, we also used strict match and relaxed match measures to compute entity recognition performance. NER strict match considers the predicted entity positive if its boundary is equal to the reference boundary. Relaxed match is measured by a variant of the Jaccard index applied to segments: the similarity of entity pairs is measured as the ratio between the size of the overlapping text segments and the size of the two merged text segments.

To avoid counting substitution errors twice, we used the Slot Error Rate (SER) that has been devised to undertake this shortcoming [48]. For entity linking evaluation, we used strict match and Wang's similarity [49] to compute hard and soft semantic annotation scores. Strict match requires that the predicted class is equal to the expected class. Wang's similarity takes into account the number of common ancestors between the predicted label and the label to be predicted with closer terms contributing more to the similarity than more distant terms.

**Rule-based methods.** The recognition of phenotypes and traits and their linking to WTO classes was achieved by the *ToMap* rule-based method [32] described in the Related Work section above, which we find suitable for accurately identifying the borders of complex syntactic structures without training.

The species have been recognized and linked to the NCBI taxonomy by the *AlvisTaxa* taxon recognition component [46] described in the Related Work section above. The results are reported in Table 5.

The prediction of *AlvisTaxa* is good with 0.85 F-measure. The analysis of false negatives shows that non-standard abbreviations such as *Th. Elongatum* were not recognized. Some were partly recognized, e.g., *Triticum aestivum* instead of *Triticum aestivum* L. Some of the incomplete entities were assigned to the wrong NCBI class, e.g., the entity *Psathyrostachys huashania* Keng was recognized as *Psathyrostachys* and then linked to the genus instead of the species *Psathyrostachys huashanica*. A preprocessing problem makes the names into parentheses unrecognizable.

The poor prediction of correct boundaries of traits and phenotypes partly explains the medium performances, for example, the prediction of *spikelet number* instead of *total spikelet number per spike*. Performance increases with boundary relaxation as measured by the Jaccard index (column 2). The very low performances of phenotype prediction type may also be due to the boundary issue. Many wrongly predicted phenotypes were predicted as traits with a substring of the correct entity. An example is the prediction of the expression *grain quality* as a trait instead of the prediction of *decreased **grain quality*** as a phenotype. A similar example is the prediction of the *protein content* trait instead of the *lower **protein content*** phenotype. We measured the performance when traits and phenotypes are confounded. The bottom row of the table (Phenotype = Trait) shows the effect. The performance increases significantly and

**Table 5. Performances of the rule-based methods *AlvisTaxa* and *ToMap* on *TaeC* for the named entity recognition and the named entity linking tasks of the species, trait, phenotype types.** Strict match measure and relaxed match measures of the entity span and of the class are shown in columns (1) and (2), respectively.

| | (1) Strict match (NER&NEL) | | | | (2) Jaccard index (NER) and Wang distance (NEL) | | | |
|---|---|---|---|---|---|---|---|---|
| | Precision | Recall | F1 | SER | Precision | Recall | F1 | SER |
| **Species (AlvisTaxa)** | 0.91 | 0.80 | 0.85 | 0.28 | 0.96 | 0.84 | 0.90 | 0.18 |
| **Trait (ToMap)** | 0.44 | 0.27 | 0.33 | 1.07 | 0.57 | 0.34 | 0.43 | 0.81 |
| **Phenotype (ToMap)** | 0.07 | 0.04 | 0.05 | 1.53 | 0.16 | 0.10 | 0.12 | 1.33 |
| **All** | 0.59 | 0.41 | 0.49 | 0.87 | 0.68 | 0.48 | 0.56 | 0.67 |
| **Phenotype = Trait** | 0.37 | 0.22 | 0.28 | 1.16 | 0.55 | 0.33 | 0.41 | 0.80 |

gets close to the trait prediction performance. This shows that the poor phenotype prediction is mainly caused by the misclassification of phenotypes as traits.

**Machine learning methods.** The machine learning methods were all trained using the train and development subsets. We evaluated their performances by 10-fold cross-validation measured by micro-average of precision, recall, and F1.

*Named entity recognition.* For the NER task evaluation, we fine-tuned RoBERTa [35] and BioBERT [37] models. Table 6 presents their precision, recall, and F1 performances, as well as the Jaccard index measure.

Both methods perform well considering the difficulty of the phenotype and trait recognition task. The recognition of species is surprisingly high, given that the algorithms do not use a species dictionary. RoBERTa achieved significantly better performances than BioBERT (in bold).

The confusion between phenotype and trait types could also explain here the very low performance of both methods for phenotype recognition. To test this hypothesis, we measure the performances of the models without distinguishing between both types. We call *Characteristics* the union of the two types. The experimental results are shown in Table 7.

The experimental results confirm our hypothesis. When traits and phenotypes are confounded into a single category, the algorithms show improved performance compared to when they are treated as separate categories. The overall performance of both methods significantly increases by + 3 points for RoBERTa and +4 points for BioBERT. It is worth noticing that not only traits and phenotypes are better recognized, but also species. These preliminary results encourage to the evaluation of the methods for the merged type in the future.

*Named entity linking.* Regarding the NEL evaluation, we assessed the performance of C-Norm [42] and BioSyn [45] for the prediction of trait and phenotype classes. We did not

**Table 6. Performance evaluation of RoBERTa and BioBERTa machine learning methods for the recognition of phenotype, trait, and species entities.** Performances is measured by precision, recall, micro-F1 measure and Jaccard index as a relaxed measure. The best performance of the two methods is shown in bold.

| Method | Entity | Precision | Recall | F1 | Jaccard index |
|---|---|---|---|---|---|
| RoBERTa | Overall | **0.67** | **0.33** | **0.44** | **0.50** |
| | Phenotype | **0.92** | **0.09** | **0.17** | **0.26** |
| | Trait | 0.58 | **0.31** | **0.41** | **0.47** |
| | Species | **0.79** | **0.44** | **0.57** | **0.62** |
| BioBERT | Overall | 0.61 | 0.29 | 0.39 | 0.46 |
| | Phenotype | 0.84 | 0.03 | 0.05 | 0.15 |
| | Trait | **0.6** | 0.29 | 0.39 | 0.44 |
| | Species | 0.62 | 0.39 | 0.48 | 0.54 |

**Table 7. Performance evaluation of RoBERTa and BioBERTa methods for the recognition of characteristics and species entities.** Performances is measured by precision, recall, micro-F1 measure and Jaccard index as a relaxed measure. The best performance of the two methods is shown in bold.

| Method | Entity | Precision | Recall | F1 | Jaccard index |
|---|---|---|---|---|---|
| RoBERTa | Overall | **0.71** | **0.35** | **0.47** | **0.51** |
| | Characteristics | **0.67** | **0.31** | **0.42** | **0.49** |
| | Species | **0.77** | **0.44** | **0.56** | **0.62** |
| BioBERT | Overall | 0.63 | 0.32 | 0.43 | 0.47 |
| | Characteristics | 0.61 | 0.28 | 0.38 | 0.46 |
| | Species | 0.66 | 0.4 | 0.50 | 0.58 |

**Table 8. Performance evaluation of BioSyn and C-Norm NEL methods on the prediction of phenotype and trait class from entities, i.e. entity linking.** The performance is measured by precision, recall, micro-F1 measure and Wang similarity as a relaxed measure. The best performance of the two methods is shown in bold.

| Method | Entity | Precision | Recall | F1 | Wang similarity |
|---|---|---|---|---|---|
| C-Norm | Overall | 0.83 | 0.81 | **0.82** | **0.91** |
| | Phenotype | 0.88 | 0.82 | 0.85 | 0.87 |
| | Trait | 0.84 | 0.82 | **0.83** | **0.92** |
| BioSyn | Overall | 0.84 | 0.82 | 0.83 | 0.90 |
| | Phenotype | 0.92 | 0.87 | **0.89** | **0.89** |
| | Trait | 0.84 | 0.81 | 0.82 | 0.9 |

**Table 9. Performance of combined NER and NEL methods RoBERTa + C-Norm on phenotype and trait entities.** Performances is measured by micro-F1 measure and Wang similarity as a relaxed measure.

| Entity | F1-micro | Wang similarity |
|---|---|---|
| Phenotype | 0.03 | 0.21 |
| Trait | 0.50 | 0.78 |
| Overall | 0.49 | 0.77 |
| Overall (Phenotype ∪ Trait) | 0.55 | 0.82 |

evaluate the prediction of species because the size of the NCBI taxonomy is beyond the capacity of the algorithms. The two models were provided with the gold-named entity for training and inference. The results are reported in Table 8.

Both methods perform similarly on trait and phenotype types and achieved good scores compared to similar biomedical normalization tasks such as Bacteria Biotope'19 BB-Norm [19].

*Named entity recognition and linking.* The previous experiments to infer the classes to be linked to the entities were done using the gold reference entities. In this section, we present the results obtained by linking the predicted entities to the class, that is, successively using the best method RoBERTa for NER and applying the best method C-Norm to RoBERTa results for NEL. The performances obtained are displayed in Table 9.

As expected, the final scores decreased compared to NEL applied to gold entities. The bad recognition of the phenotype entities has a great impact with an F1 score equal to 0.03; however, the Wang similarity at 0.21 for phenotypes shows that the predicted classes are not correct, but a significant part of them belong to the ancestors of the class to predict. The same comment can be made more generally by comparing the F1 scores and the Wang similarity measures that differ by almost 30 points. These encouraging results lead us to conclude that there is considerable room for improvement in using more effective and better-adapted methods.

## Discussion

In this paper, we have presented the rationale for the construction of the new reference corpus *TaeC* of 528 documents relevant to wheat breeding research, and more generally to cultivated plant phenotyping. *TaeC* is annotated with three entity types, i.e., traits, phenotypes, and species, and each entity is linked to a reference class from NCBI taxonomy for species and WTO for traits and phenotypes, yielding 10,635 occurrences of 340 classes.

We showed that the traits and phenotypes mentions are frequently expressed by complex terms and that the labels of the ontology classes and the textual expressions differ in many ways. Beyond the usual linguistic phenomena (e.g. variations in morphology and syntax, synonymy), paraphrasing and causal inference involve more complex mappings between text and class labels and deeper expertise in the wheat domain. This prevents the straightforward application of string matching for NER and NEL. This observation applies to other ontologies than WTO such as CO_321 and more generally to Crop Ontologies. Our corpus analysis suggests that this is not limited to wheat but could be generalized to cultivated plants. The treatment of such linguistic phenomena may be beyond the capabilities of usual entity recognition and linking tools. More powerful information extraction methods are therefore needed to automate the recognition of the trait and phenotype entities in the text and link them to the corresponding classes in ontologies.

Our experimental results with baseline rule-based and LLMs show that *TaeC* is useful for training and evaluating NER and NEL methods–the performances are similar to those obtained with other corpora on microbial and human phenotypes–but there is considerable room for improvement in using more effective and better-adapted methods.

We have considered extending the annotation schema to other entities and relationships as in [13]. The genes and varieties and their relationships with traits and phenotypes are an annotation priority for studying of the genotype-phenotype relationship. We annotated a document sample to estimate the complexity of the task and the expertise required. The abundance of gene transfers and mutations and the complexity of the cultivar obtention through breeding make the descriptions dense and convoluted. Figuring out how entities are related across multiple events and across sentences is a new challenge that requires deep expertise.

We have taken a close look at the naming of wheat varieties and cultivars. First, wheat varieties may be referred to by their commercial names (e.g., *European Plant variety database*; https://ec.europa.eu/food/plant-variety-portal/), which make them easy to recognize. However, they represent only a small proportion of the mentions. More frequent and valuable for the design of a gold corpus are mentions of cultivars or accessions obtained by genetic modification or crossing. Unfortunately, they are denoted by more complex expressions, including verbal ones. Table 10 shows four illustrative examples of such mentions of cultivars.

Their manual annotation therefore requires a high level of expertise and resources including for defining exactly what a cultivar is along a continuous line from the commercial variety

**Table 10. Examples names and descriptions of cultivars from literature illustrate the complexity of their annotation.**

In this study, **$F_2$ and $F_{2:3}$ lines derived from the cross 10-A × BE89** were used to construct a genetic map [PMID: 32549690]

we produced a **F(6:7) recombinant inbred line (RIL) population by crossing Wangshuibai with the scab-susceptible cultivar Nanda2419**. [PMID: 15290053]

Moreover, **wheat cultivars with Pina-null/Pinb-null allele** have significantly higher SKCS hardness index [..] [PMID: 24011219]

The **wheat accession H9020-1-6-8-3 is a translocation line previously developed from interspecific hybridization between wheat genotype 7182 and *Psathyrostachys huashanica***. [PMID: 30727301]

name to the modified genotype description. We keep the plan of annotating varieties, genes and markers in our long-term future work, to address the need to bridge the phenotypic and genetic data for integrated marker-assisted plant selection.

In medium term work, we will train up-to-date NER/NEL LLMs with *TaeC* and integrate the methods into the wheat AlvisNLP workflow with the aim of automatically annotating PubMed references on wheat and using it as a service of WheatIS-Faidare.

## Conclusion

Textual sources are complex to parse for feeding linked knowledge-based systems in the agronomic domain. Recent advances in deep learning and transformer-based methods are appealing for their application to the crop domain, especially to contribute to complex biological and agronomic questions related to the role of genes in expressing desirable phenotypes such as drought tolerance or resistance to diseases. The adaptation and the application of these machine learning methods require annotated corpora for their training and evaluation.

For this purpose, *Taec* represents a unique and original corpus, firstly because of its focus on cultivated plants, and secondly because of the nature of the NER and NEL tasks that it allows us to study. In domain-specific NLP, many types of entities are denoted by complex expressions that differ from the labels of the reference ontologies as is the case here. In addition, domain-specific annotated corpora with ontologies rich in hierarchical or non-hierarchical relationships, such as *Taec* and WTO are rare outside the human health domain and provide a new area for NEL methods to explore.

## Acknowledgments

The authors thank Léonard Zweigenbaum (INRAE) for his contribution to the annotation and, Thierry Marcel and Jacques Le Gouis of INRAE for their contribution to the technical questions relating to the description of the phenotyping of wheat.

The authors thank the Migale platform for providing the resources to run AlvisNLP services (MIGALE, INRAE, 2020. Migale Bioinformatics Facility, doi: 10.15454/1. 5572390655343293E12).

## Author Contributions

**Conceptualization:** Claire Nédellec, Robert Bossy.

**Formal analysis:** Robert Bossy.

**Funding acquisition:** Claire Nédellec.

**Methodology:** Robert Bossy, Mariya Borovikova, Louise Deléger.

**Project administration:** Claire Nédellec.

**Resources:** Claire Nédellec, Clara Sauvion, Robert Bossy, Louise Deléger.

**Software:** Robert Bossy, Mariya Borovikova, Louise Deléger.

**Supervision:** Claire Nédellec.

**Validation:** Claire Nédellec, Clara Sauvion, Robert Bossy, Mariya Borovikova.

**Writing – original draft:** Claire Nédellec.

**Writing – review & editing:** Claire Nédellec, Clara Sauvion, Robert Bossy, Mariya Borovikova, Louise Deléger.

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
