## [Decision Letter · Decision Letter 0]

16 Jan 2024

PONE-D-23-38159Taec: a Manually annotated text dataset for trait and phenotype extraction and entity linking in wheat breeding literature

Dear Dr. Nédellec,

Thank you for submitting your manuscript to PLOS ONE. After careful consideration, we feel that it has merit but does not fully meet PLOS ONE’s publication criteria as it currently stands. Therefore, we invite you to submit a revised version of the manuscript that addresses the points raised during the review process.

The comments from reviewer 1 and reviewer 2 plus the academic editor are indicated at the end of this letter. 

Please submit your revised manuscript by Mar 01 2024 11:59PM.  If you will need more time than this to complete your revisions, please reply to this message or contact the journal office at plosone@plos.org. Please include the following items when submitting your revised manuscript:

We look forward to receiving your revised manuscript.

Kind regards,

Wuletaw Tadesse

Academic Editor

PLOS ONE

Journal Requirements:

Additional Editor Comments:

The manuscript addresses on of the important issues of information extraction and utilization. There are many publications in wheat both at phenotypic and molecular levels. However, there are limited tools for proper annotation and utilization. The authors of Taec have shown the possibility to link phenotype and gene data in wheat by using more than 540 publications. Though currently, it is done manually, it will lead to automated extraction systems.

Comments: 1. Please state the rationale in a simple and summarized ways. 2. clearly indicate the objectives. 3. Indicate the details in the way forward especially automation.

Reviewers' comments:

Reviewer's Responses to Questions

**Comments to the Author**

1. Is the manuscript technically sound, and do the data support the conclusions?

Reviewer #1: Yes

Reviewer #2: No

2. Has the statistical analysis been performed appropriately and rigorously? 

Reviewer #1: Yes

Reviewer #2: N/A

3. Have the authors made all data underlying the findings in their manuscript fully available?

Reviewer #1: Yes

Reviewer #2: Yes

4. Is the manuscript presented in an intelligible fashion and written in standard English?

Reviewer #1: No

Reviewer #2: Yes

5. Review Comments to the Author

Reviewer #1: In this manuscript the authors are addressing the very valid, and complex, challenge of using text analytics to make the scientific literature of wheat more accessible. Text analytics have been successfully used in other fields, such as biomedicine and law, to automate the synthesis of information from natural language. They point out that no specific corpus, the first step for successful language analytics, has been developed for wheat research. In this manuscript the authors describe the development of such a corpus. They furthermore used the corpus with a variety of language models and present the statistical comparisons of the various models. They conclude that their corpus is suitable for the training of language models. The presented work is relevant for the emerging field of automated language recognition in the biological sciences, especially interesting is the focus on plant/agricultural crops as these have been neglected in the past. However, I have a number of major and minor concerns, which would need to be addressed before further consideration for publication:

Major revisions:

1. I believe the study is not properly placed into the context of previous publications and has failed to acknowledge highly relevant studies. The aim of the authors was to use text analytics to make the large amount of literature on genotype-phenotype relations in wheat more accessible and searchable. I believe this goal is very much at the heart of “knetminer”, which intertwines vast amounts of data from a spectrum of species, to bridge the gap between the considerable amount of available information from publications and scientific discovery. Knetminer has also used text analytics and has already achieved a very robust information network, which is easily accessible via the web-based interphase. Furthermore a specific wheat knetminer network has been published already. Without the authors comparing their achievements to knetminer, and pointing out the potential advancements of their research in this field, I believe the current manuscript as insufficient for publication.

2. The authors state in the conclusion that “In short-term work, the NER/NEL methods we trained with Taec will be integrated into a AlvisNLP workflow that will be used as a service of the D2KAB project to automatically annotate all PubMed references on wheat.” I believe this short-term goal should be part of the presented study as otherwise an important output/result of this work is missing.

3. Overall, the manuscript is not up to standards regarding the structure and formatting of text, tables, and figures. I believe the manuscript must be overall improved before publication. Most importantly, the manuscript is missing a discussion. The current discussion (line 524-532) only describes the problem of using text analytics for wheat varieties. It does not discuss the presented results, or places these in the context of current literature. I have listed all other concerns regarding style and structure under the minor revision section.

Minor revisions:

1. As mentioned under the major revision point 3, the sections are not well structured. The introduction is not very concise and could be improved, furthermore the results and methods are not well organized, for example: From line 509, The following section is methods and should thus be moved to the relevant section, rather than appearing in results section.

2. TaeC spelling not consistent throughout text (varying between Taec/TaeC, sometimes italicized). Please correct throughout manuscript.

3. In multiple incidences words are capitalized in the middle of the sentence (incorrectly), for example line 130. Please change this throughout manuscript.

4. Table 3 has un-helpful color scheme, no heading for columns apart from column 1, unnecessary use of # before row entries, use of abbreviations with no explanation (av.), and insufficient explanation in the table heading. However, other tables suffer from some of the same issues and would need to be revised, and standardized within the manuscript.

5. The example of “earliness per se” used in figures and text throughout the manuscript might be miss-leading. It is a very simple File.txt entry, which has exactly one trait and one phenotype match in file a1. It also has very clean semantic annotations in file a2. Opening more than 10 random files of the supplemental data I could not find any case of such a clear annotation. Most file a1 contained very long, if not all parts of the text in the txt file. In file a2 multiple, species or phenotype annotations are listed for each text. I believe a more representative example could be chosen, or a section must be added to inform the reader that a simplified example was chosen.

6. One of the examples of ontology given in table 1 is not consistent with the use of the term ontology, nor the other examples: “Number of fertile florets at anthesis” is labeled as “percentage of florets without grain”. I do agree that both terms are related to the same plant structure (florets) and also describe similar phenotypes, however the number of fertile florets at anthesis is not the same as percentage of florets without grain. Anthesis happens before grain set, most florets are fertile at this stage, although not all, and most fertile florets will set grain, albeit not all. And no floret contains grains at anthesis! The percentage of florets without grain is measured later in crop development and the percentage is determined by (1) the percentage of fertile florets at anthesis and (2) the number of these florets that set grain. Could the authors explain if in table one the two items are only ontology-matches because they are related to a similar phenotype, or if indeed the pairs should be synonymous to each other, in which case the floret example would have to be removed.

Reviewer #2: The paper addresses the need for a specialized corpus in wheat breeding research, providing valuable annotations for traits, phenotypes, and species. TaeC serves as a resource for training and evaluating NER/NEL methods, contributing to advancements in cultivated plant phenotyping. The paper's approach aligns with broader initiatives aiming to integrate textual annotations and crop ontology data for comprehensive knowledge bases in wheat. However, some points are already drawn by the authors which are considered as limitations to the comprehensive scope and applicability of the corpus: TaeC currently lacks annotations for varieties and cultivars, a notable limitation given their significance in wheat breeding. The absence of these entities may restrict the corpus's applicability in capturing the full spectrum of wheat-related research, particularly in contexts where specific varieties or cultivars play a crucial role. Additionally, an inherent bottleneck lies in the intricate nature of annotating complex expressions related to cultivars or accessions. The manual annotation process demands a high level of expertise and resources, making it labor-intensive and potentially limiting scalability. Addressing this bottleneck requires exploring automated or semi-automated annotation approaches to enhance efficiency. While TaeC excels in annotating traits, phenotypes, and species, there's a need for explicit discussion on the corpus's scope. Clearly defining the intended applications and potential constraints ensures a better understanding of TaeC's generalizability and use cases. This clarification is vital for researchers and practitioners leveraging the corpus for various applications within the field of cultivated plant phenotyping.

6. PLOS authors have the option to publish the peer review history of their article (what does this mean?). If published, this will include your full peer review and any attached files.

Reviewer #1: No

Reviewer #2: No

---

## [Author Response · Author response to Decision Letter 0]

17 Apr 2024

The editor's comments were: Comments: 1. Please state the rationale in a simple and summarized ways. 2. clearly indicate the objectives. 3. Indicate the details in the way forward especially automation." 

Comments (1) and (2) have been addressed by completely rewriting the Abstract and Introduction sections and moving some paragraphs to the next section. We have clarified how this research is related to automation to answer Comment (3). This research is about a corpus to be used for the training and evaluation of automatic methods. We have added descriptions of the automatic systems that would benefit from using the corpus. The way we responded to these comments is detailed in the "Response to Reviewers" document.

---

## [Decision Letter · Decision Letter 1]

31 May 2024

Taec: a manually annotated text dataset for trait and phenotype extraction and entity linking in wheat breeding literature

PONE-D-23-38159R1

Dear Dr. Claire Nédellec

We’re pleased to inform you that your manuscript has been judged scientifically suitable for publication and will be formally accepted for publication once it meets all outstanding technical requirements.

Kind regards,

Wuletaw Tadesse

Academic Editor

PLOS ONE

Additional Editor Comments (optional):

No further comments. The manuscript is accepted for publication in its current form.

Reviewers' comments:

Reviewer's Responses to Questions

**Comments to the Author**

1. If the authors have adequately addressed your comments raised in a previous round of review and you feel that this manuscript is now acceptable for publication, you may indicate that here to bypass the “Comments to the Author” section, enter your conflict of interest statement in the “Confidential to Editor” section, and submit your "Accept" recommendation.

Reviewer #1: All comments have been addressed

Reviewer #2: All comments have been addressed

2. Is the manuscript technically sound, and do the data support the conclusions?

Reviewer #1: (No Response)

Reviewer #2: Yes

3. Has the statistical analysis been performed appropriately and rigorously? 

Reviewer #1: (No Response)

Reviewer #2: N/A

4. Have the authors made all data underlying the findings in their manuscript fully available?

Reviewer #1: (No Response)

Reviewer #2: Yes

5. Is the manuscript presented in an intelligible fashion and written in standard English?

Reviewer #1: (No Response)

Reviewer #2: Yes

6. Review Comments to the Author

Reviewer #1: (No Response)

Reviewer #2: The manuscript has been significantly improved in response to the feedback provided. The authors effectively addressed the lack of annotations for varieties and cultivars by acknowledging the challenges involved. They explained the complexity of annotating these entities due to diverse linguistic expressions and the labor-intensive nature of manual annotation. They also explored automated and semi-automated methods using commercial accession catalogues and the GRIS database, but these yielded noisy and incomplete results. Additionally, the authors clarified the scope of the TaeC corpus, its intended applications, and potential constraints. They made enhancements in the abstract, introduction, and discussion, particularly highlighting the benefits for various wheat information networks. These clarifications may help researchers and practitioners understand the corpus's generalizability and specific use cases within cultivated plant phenotyping. With these comprehensive responses and improvements, the manuscript is now more robust and informative, addressing the initial concerns effectively, and thus, I do not suggest any further revisions.

7. PLOS authors have the option to publish the peer review history of their article (what does this mean?). If published, this will include your full peer review and any attached files.

Reviewer #1: No

Reviewer #2: **Yes: **Samira El Hanafi

---

## [Editor Report · Acceptance letter]

4 Jun 2024

PONE-D-23-38159R1 

PLOS ONE

Dear Dr. Nédellec, 

I'm pleased to inform you that your manuscript has been deemed suitable for publication in PLOS ONE. Congratulations! Your manuscript is now being handed over to our production team.

Kind regards, 

on behalf of

Dr. Wuletaw Tadesse 

Academic Editor

PLOS ONE